# Cosmogenic nuclide-derived downcutting rates of canyons within large limestone plateaus of southern Massif Central (France) reveal a different regional speleogenesis of karst networks

Oswald Malcles[1], Philippe Vernant[1], David Fink[2], Gaël Cazes[1], Jean-François Ritz[1], Toshiyuki Fujioka[3], Jean Chéry[1]

[1]Géosciences Montpellier, Université de Montpellier, CNRS, Montpellier, France
[2]Australian Nuclear Science and Technology Organisation, Sydney, Australia
[3]Centro Nacional de Investigación sobre la Evolución Humana (CENIEH), Burgos, Spain

*Correspondence to*: Oswald Malcles (oswald.malcles@umontpellier.fr)

**Abstract.** We present 35 new burial ages (27 sites) based on $^{26}Al/^{10}Be$ ratios of terrestrial cosmogenic radionuclides measured in clasts and sediments deep within 12 caves in the southern Massif Central, France. Our results together with previously published burial ages, verifies that cave morphogenesis has been continuously active in this region for at least the past ~6 Myrs. Combining sample burial ages with their associated cave elevation above modern stream bed gives a mean regional incision rate of 88 ± 5 m/Ma for the Grands Causses area. South of the Cevennes Fault Zone bordering the Grands Causses, the incision rate is 43 ± 5 m/Ma, suggesting that this difference might be accommodated by the fault zone. Sediment burial ages from caves which are not located on river valley flanks or cliff walls are surprisingly too young compared to their expected ages when calculated using this regional average river incision rate. This suggests that the classical epigenic speleogenesis model that presumes a direct correlation between cave level development and regional base level lowering does not apply for the study area. Therefore, we propose that regional speleogenesis is mainly controlled by removal of ghost-rocks by headward erosion from river canyons to central parts of the plateaus, emptying incipient primokarst passages to create cave systems. Our results suggest a continuum process from hypogene primokarst composed of passages filled with ghost-rock to one of epigene karst dynamics emptying these passages and creating cave networks. We propose these processes to be the major mechanism in the southern Massif Central that initiates speleogenesis and controls the geometry of the networks. In this region tiered karst cannot be associated with the pace of incision of the major rivers but must be explained by former ghost-rock (or hypogene) processes.

**1 Introduction – the origin of caves**

Speleogenesis has been an ongoing research topic for decades and debate on the spatial and temporal evolution of caves is equally as old (Palmer, 2017). The current paradigm of epigenic speleogenesis (Fig. 1) includes: (1) steep vadose upstream sections converging into (2) phreatic or epiphreatic sub-horizontal passages constrained by the local water table and (3) subsequent groundwater emergence as springs at river valley floor (e.g. Ford and Williams, 2007, Audra and Palmer, 2013, Harmand et al., 2017). In this epigenic model, the sub-horizontal passages, called cave levels or tiered karst, are assumed to have been created by dissolution of bedrock during a prolonged period of base-level river stability. When river incision resumes, both the contemporary base-level and water table lower, allowing formation of new passages while the previous generation of cave levels (usually higher in elevation) is isolated from further fluvial occupation. Therefore, each cave level is considered to reflect a base level at a certain period of time in the past. This broadly accepted correlation between elevation of successive horizontal cave passages and river valley evolution is commonly used to study speleogenesis and quantify incision rates (e.g., Granger et al 1997, Granger et al 2001, Stock et al., 2005, Haeuselmann et al., 2007, Harmand et al., 2017). The complication with this simple view of the epigenic speleogenesis paradigm (ESP) is that although groundwater discharge dissolves carbonates, it also simultaneously physically erodes and transports insoluble residues from bedrock sections. This process implies that large enough passages must exist prior to speleogenesis onset of sub-horizonal cave levels to avoid clogging of passages by insoluble bedrock fragments. To get around this problem, the conventional model implicitly assumes that, for most of the time, the open fractures in bedrock allow the removal of soluble and insoluble products at the same time facilitating speleogenesis (in depth discussions about the implications of the chemical weathering and mechanical erosion processes can be found in Dubois et al. (2014) or Quinif (2010) for example). Other models have been proposed to explain speleogenesis, but they are commonly viewed as marginal processes "because these types of speleogenesis are not connected to a fluvial base level" Harmand et al. (2017). They include hypogenic cave formation mainly due to confined deep groundwater with a dissolution potential not related to surface processes (e.g., Klimchouk, 2012) or ghost-rock karstification (e.g., Dubois et al. 2014). Ghost-rock karstogenesis (also called phantomization) has been first described by Schmidt (1974) but mostly overlooked as a major karstification process and, according to Klimchouk (2017), it is a specific manifestation of hypogenic karstification. For others (e.g., Quinif, 2010, Dubois et al. 2014, Rodet, 2014) phantomization can be a major regional karstification process involving a first stage of bedrock chemical weathering along least flow resistance paths (faults, fractures, bedding planes), with subsequent removal of the soluble matrix under low hydrodynamic conditions leaving the rock structure with the more resistant insoluble matrix essentially preserved. During the first phase which is limited to chemical weathering, only incipient passages are formed along weak flow paths (i.e., 'ghost-like' karstification) though often they can be misinterpreted as cave sediment deposits. The progressive alteration of the rock – the ghost weathering process – leads to interconnection of ghost-rock zones. This network of connected ghost-rock zones, which Rodet (2014) defines as "primokarst", is the incipient geometry along which cave networks will eventually develop depending on hydrodynamic conditions during the speleogenesis phase. Indeed, if hydrodynamic

conditions change, allowing rapid water flow, mechanical erosion of the ghost-rock will then preferentially open these weaker pre-existing paths and create caves.

Whatever model of karstification is chosen, the passage formation is the result of 3 steps (e.g., Klimchouk, 2015): (1) the early stage corresponds to widening of the flow path-ways or primokarst formation; (2) the breakthrough phase which can be seen as the formation of efficient passages where the water can flow quicky and easily; (3) the last phase is when the main drains are well established allowing the stabilization of the system and the growth of the principal conduits. The main difference between the epigene karstification and the other processes is the relation to the regional base level. Epigene karst geometry is directly related to river incision dynamic, while hypogenic and ghost-rock karstification occur below the base level and subsequent tiered karst geometries cannot be interpreted in terms of river entrenchment phases. Indeed, the ghost-rock formation is considered to occur principally below the base level with a low energy water flow exporting soluble elements without exporting the insolubles. This period of localized alteration creates geometries including horizontal primokarst passages. When the primokarst passages are ripped open by the valley entrenchment, water can freely flow through the newly formed opening allowing hydrodynamic conditions to change for high energy and export ghost-rocks while emptying the primokarst network and forming cave passages with eventual horizontal galleries. This model does not need river level steadiness. We assume that most of the time the intersection of the primokarst by the topography is related to the entrenchment of the river but other processes, as escarpment recession for example, could also provide a physically satisfactory explanation. The process of headward ghost-rock drain and subsequent cave creation has been observed in real-time in Belgian quarries (e.g. Quinif, 2010, Dubois et al. 2014). Erosion of the ghost-rock can occur below the base level as long as the hydrological gradient is sufficient to create a high enough energy water flow to permit the export of the insolubles. Large water flow loops at depth have been proposed to explain some hypogene cases since the flow is upward on one end of the loop (Klimchouck, 2017). Dandurand et al. (2019) refer to a similar process with large convection cells of water at depth to explain ghost-rock formation and its subsequent drain, sometimes creating a deep sump at more than 100m below the base level. In this model, deep convective cells are proposed as a satisfactory explanation for primokarst formation, subsequent drain, and finally deep phreatic loops such as Fontaine de Vaucluse or Touvre spring.

In this study, we investigate speleogenesis of the Grands Causses region, southern Massif Central, France (Fig. 2). We apply the pioneering methods of Granger et al. (1997) using terrestrial cosmogenic radionuclide (TCN) $^{26}$Al/$^{10}$Be ratios to estimate burial ages of quartz rich sediment and quartz cobble cave infill together with detail cave mapping to quantify river incision rates. We spatially distinguish burial ages between caves opened on river canyon walls to those centrally located in plateaus to test the above models. Our results challenge the pervasive current ESP model of speleogenesis.

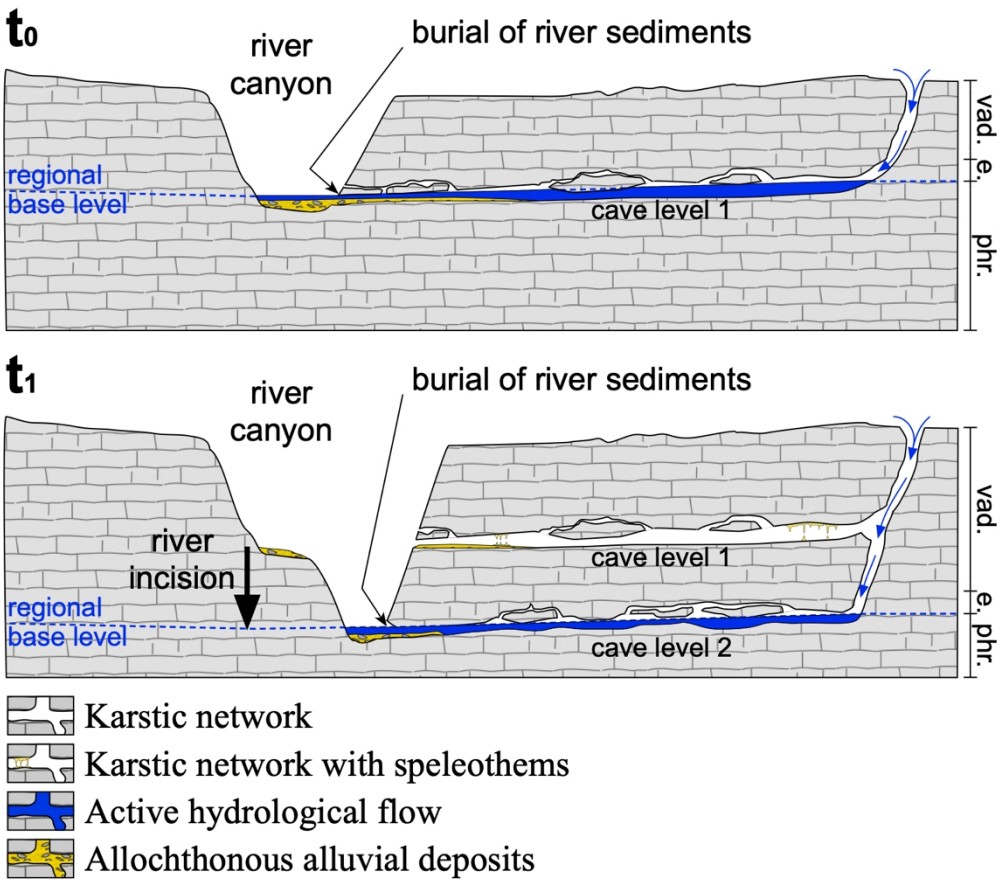

**Figure 1: Cave level development accordingly to the commonly accepted epiphreatic speleogenesis paradigm (ESP). Fig A (at t = t0): Water entering from the plateau dissolves and creates steep passages in the vadose zone (vad.) to connect to the epiphreatic (e.) and phreatic (phr.) zones where it forms sub-horizontal passages linked to the regional base level. Fig B (at t = t1): subsequent river incision lowers the water table creating new cave levels (level 2) at the regional base level. Previously formed (older) cave**
**levels (level 1) become abandoned.**

## 2 Geology and Plio-Quaternary geomorphologic evolution of the Grands Causses

The Grands Causses region (Fig. 2) is a large, elevated plateau of thick sub-horizontal Mesozoic carbonated series overlying a Hercynian metamorphic and plutonic crystalline basement. Mean surface elevation is around 800m above sea level (a.s.l.) and its south-east margin is defined by a steep slope along the Cevennes Fault Zone (CFZ). The latest activity of this
Hercynian-inherited major fault system according to Seranne et al. (2002) is an uplift of the north-west sector during the Serravalien/Tortonien (prior to ca. 8 Ma). Several rivers have their upper riverbeds and sources within crystalline areas (granite and schists). Their lower riverbeds carve deeply into the limestones on their journey to the Mediterranean Sea or the Atlantic Ocean sculpting canyons that can be up to 400m deep (Fig. 2). Incision rates and timing of canyon formation are

still debated. Since the early 2000's it was generally considered that the Grands Causses morphology was mostly inherited from the Miocene without significant incision later during the Quaternary (Seranne et al., 2002). Recent quantification of incision rates based on TCN burial dating in caves of the Rieutord river yielded rates of ~ 80 m/Ma over the last ~2 Ma (Malcles et al., 2020a) on the Mediterranean side and 40 to 120 m/Ma for the Jonte River on the Atlantic side over the last ~8 Ma (Sartégou et al. 2018).

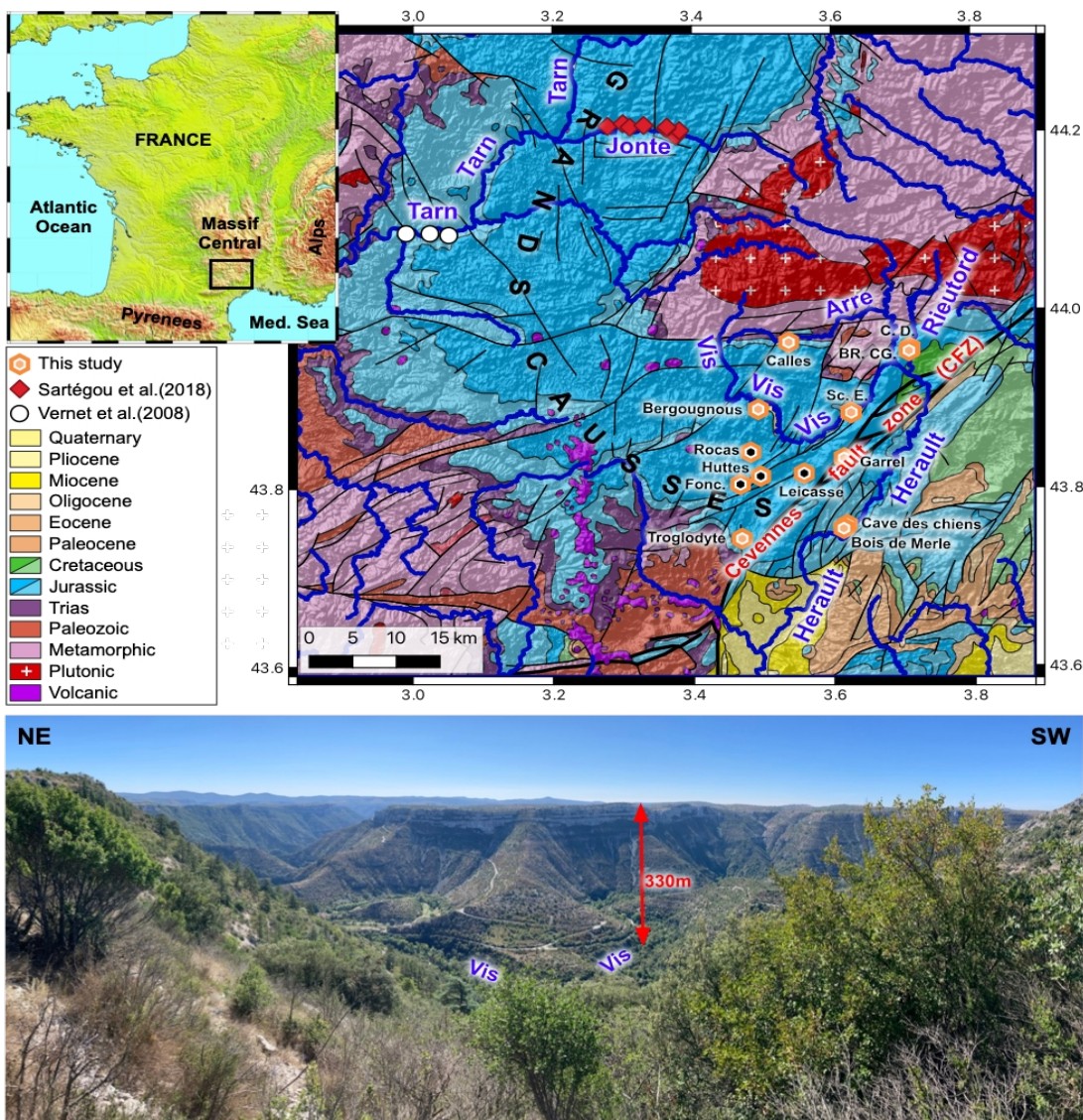

**Figure 2: Top: Simplified geological map of the study area and sample locations. The metamorphic and plutonic bedrock provide quartz rich sediments to the Tarn, Jonte, Arre, Rieutord, Vis and Herault rivers. For the Jurassic and Cretaceous rocks, darker colors indicate thick limestone formations while lighter colors are for marls and thin limestone formations. Symbols with black hexagons (Fonc, Huttes, Rocas, and Leicasse) indicate sampled caves located away from the river canyons within a central section of the plateau. Sc.: Scorpions, E.: Escoutet, BR.: Bord de Route, CG.: Camp de Guerre, C.: Cuillere, D.: Dugou, Fonc.: Fonctionnaire. Bottom: Photograph of the Vis river canyon located next to the Bergougnous sample.**

# 3 Methods

## 3.1 Terrestrial Cosmogenic Nuclide (TCN) data

We use TCN burial ages to quantify regional incision rates. The method (Granger et al., 1997) is based on the change in the initial $^{26}Al/^{10}Be$ ratio produced by cosmic ray bombardment of subaerially exposed rocks, whereby after erosion and via fluvial transport, the irradiated quartz rich grains or cobbles are deposited and stored in cave systems. If burial is at a sufficient depth (such as is the case in our study) production ceases and the measured $^{26}Al/^{10}Be$ sample ratio is reduced compared to its initial ratio due to differential decay of the shorter lived $^{26}Al$ (half-time of 1.387 and 0.705 Ma for $^{10}Be$ and $^{26}Al$ respectively; Korschineck (2010); Chmeleff (2010)). We sampled quartz-rich alluvium and small cobbles in 8 new caves and resampled 4 from caves previously reported (Malcles et al., 2020a, 2020b) for a total of 35 samples. In order to provide a strong constraint in calculating river incision rates, we sampled where possible tiered caves that show horizontal galleries. The selection of caves to sample was made based on morphological evidence, as for example in the Scorpions caves that shows all the indices of being an endokarstic loop, and by published work (Camus, 2003). 3D cave topography can be obtained from the KARST3D database (KARST3D, 2019). The inventory of river canyons in this work includes 3 new Mediterranean Sea tributaries (The Hérault, Arre and Vis rivers) and a fourth canyon, the Rieutord, which was resampled (Malcles et al 2020a).

The samples were crushed, sieved, and processed with several selective chemical dissolutions to obtain pure quartz (Khol and Nishiizumi, 1992; Child et al., 2000). After final HF etchings, the samples were dissolved in full strength HF with addition of ~ 250 μg of $^9Be$ from a Be carrier solution derived from beryl mineral and assayed via ICP-MS to +/-1% in concentration. Be and Al were then separated by ion exchange chromatography and selective pH precipitations. Final BeO and $Al_2O_3$ powders were mixed with Nb and Ag, respectively, and measured, using the SIRIUS Accelerator Mass Spectrometer facility at ANSTO, Sydney Australia (Wilcken et al., 2019). All AMS results in this study were normalized to standards KN-5-4 and KN-4-4 for Be and Al, respectively (Nishiizumi et al., 2007) and corrected for background using the set of procedural chemistry blank samples prepared in each batch of 10 samples. Final uncertainties for $^{10}Be$ and $^{26}Al$ concentrations include AMS statistics, 2% (Be) and 3% (Al) standard reproducibility, 1% uncertainty in the Be carrier solution concentration, and a representative 4% uncertainty in the natural Al measurement made by inductively coupled plasma optical emission spectrometry (ICP-OES), in quadrature.

All sample identification, location, elevation (relative to modern base level), with their $^{10}Be$ and $^{26}Al$ concentrations and associated fully propagated analytical errors are given in supplementary material (Table S1). Two samples were repeated as a check on internal consistency in processing and AMS measurement.

## 3.2 Burial age modelling and paleo erosion rates

When large enough cobbles were available (> ~ 100 g), we independently processed them to obtain several burial ages for the same alluvium layer (Scorpions, Escoutet and Leicasse caves). An alternate approach is to use the isochron method (Balco and Rovey, 2008) which is, usually, expected to provide a more reliable age determination as it can accommodate the variability in pre-burial exposure history of cobbles. This method allows the removal of differences in the initial or inherited $^{10}$Be and $^{26}$Al concentrations (which result from variations in the erosional equilibrium conditions of source bedrock etc.) from the final measured cosmogenic nuclide inventories with the *a priori* assumption that post burial production was the same for all isochron samples. This method is valid as long as all the measured samples had maintained the same depths below the surface immediately following deposition in the cave system. In other cases, where only smaller elements could be found in the same deposit, we decided to process the samples as amalgamates with at least ~ 200 g of quartz at the beginning of the treatment. The latter approach provides an average concentration, hence an average burial age.

Details on burial dating theory are given in several studies (e.g., Granger et al., 1997, Granger and Muzikar, 2001, Dunai, 2010). We performed a two-step grid search to find the combination of burial ages and paleo-denudation rates (obtained based on a SLHL $^{26}$Al/$^{10}$Be production rate ratio of 6.61 with spallation SLHL production rate of 4.47 and 30.29 atm g$^{-1}$ yr$^{-1}$ for $^{10}$Be and $^{26}$Al respectively)) consistent with the measured $^{10}$Be and $^{26}$Al concentrations. In the first step we used a loose grid with burial ages ranging from 10 kyrs to 10 Myrs with 1000 values spaced evenly on a log scale and paleo-denudation rates from 0.1 to 1000 m/Ma with 200 values also spaced evenly on a log scale. We check if the obtained values for $^{10}$Be are consistent with $^{26}$Al, if not, the concentrations are considered inconsistent with a simple history of bedrock erosion followed by the river transport and finally the burial in a cave. In this case no burial age can be estimated. If a consistent set of values exists, then we perform a second grid search, similar to the first one but with tighter intervals to compute the consistent set of burial ages and paleo-denudation rates. To compute the theoretical concentrations, we account for variability in the cosmic-ray flux as a function of elevation and latitude and therefore the cosmogenic nuclide production using scaling factors. Theses scaling factors use the sampling site parameters (e.g. elevation). For the neutron spallation contribution to production, we use the Lal (1991) scaling factors. For the muon contribution, we do not use slow and fast muon production rate scaling factors (as per Braucher et al. 2013), but rather use the simpler geographic scaling method as described in Balco (2017). The best combination of burial age and paleo-denudation rate is the one leading to the smallest residual (i.e., the difference) between the measured and computed $^{10}$Be and $^{26}$Al concentrations, we do so by using the lowest chi square value of the residuals computed for all parameters of the grid. The obtained values are plotted on Figure 3 and given in supplementary material (Table S1). Both the minimal and maximal combination of burial age and erosion rate providing modeled concentrations in the range of the measured one (± 1σ) are computed to estimate uncertainties. The upper uncertainty and the lower uncertainty are the distance between the best estimation and the maximal acceptable age and erosion rate or the minimal acceptable age and erosion rate.

## 4. Results and discussion

In Figure 3 we present the burial ages from this study plotted against the cave elevation for each burial sample relative to local base level. Figure 3 also includes paired burial age–elevation pairs from previous TCN studies of Sartégou et al., (2018), for the Jonte and Tarn Rivers, from Malcles et al., (2020a) for Rieutord and OSL results for terraces of the Tarn River from Vernet et al. (2008). For cave samples directly associated with the flanks of river canyons, the elevation is relative to the modern river channel, whilst for cave samples within central plateau regions, the nearest river defines the relative elevation of the sample compared to local base level.

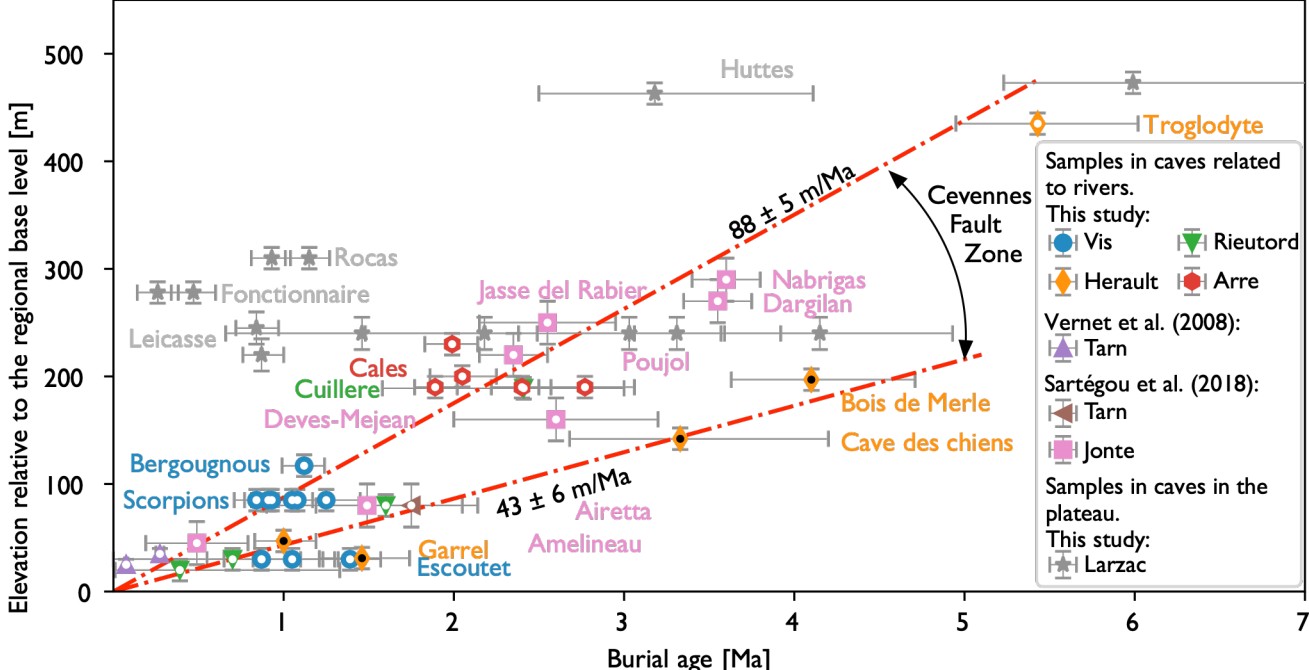

**Figure 3: Burial age vs. relative elevation to the local water table level for each sample. Grey stars indicate caves that are not located on the flanks of river channel but rather in central plateau areas, all the other caves are adjacent to or on river flanks or canyon walls. Jonte S. and Tarn S. are $^{26}$Al/$^{10}$Be results from Sartégou et al. (2018); Tarn V. are OSL results for Tarn River terraces from Vernet et al. (2008) and Rieutord are $^{26}$Al/$^{10}$Be results from Malcles 2020a. Average incision rate NW of the CFZ is calculated using samples represented by symbols with white circles inside while the rate at the SE side of the CFZ is given by the data obtained at locations represented by symbols with black circles inside.**

## 4.1 Burial ages

The new suite of burial ages spans the time range of the method from a few hundreds of thousands of years (e.g. Fonctionnaire cave) to slightly more than 5 Myrs (Troglodyte, Hutte caves). Three different populations of ages are obtained (Fig. 3) which are spatially correlated but do not obey a common age-elevation relationship.

First, the 3 cave systems associated with the Vis River canyon (Scorpions, Bergougnous and Escoutet) provide relatively young ages ~ ranging from 0.8 to 1.3 Myr, while being located no more than 120 m above the local river level. The results for these samples present, at first order, the same age-elevation relationship for river incision of about 85 meters per million years that was previously published for the caves of the Rieutord river canyon (Malcles et al., 2020a). This age-elevation relationship is also obtained for the slightly older samples from the Calles cave (amalgams) that range from 1.9 to 2.8 My and are located ~ 200 m above the Arre riverbed. Finally, the ~ 5.4 Myr and + 435 m of the Troglodyte cave deposit is also in very good agreement with this relationship. For some of these caves, multiple cobbles were processed for the same site. Isochron and amalgamate analyses were performed for Scorpions and Escoutet, and only amalgamate samples for Calles and Bergougnous. We suggest that the youngest burial age from the deposit is expected to be the best estimate of the timing before the cave becomes isolated or disconnected from further fluvial occupation due to the river entrenchment. This item is further discussed in section 4.2.

Second, samples from Garrel, Cave des chiens and Bois de Merle, which are shown as black filled data symbols in Fig 3, have burial ages that are too old with respect to their given elevation when compared to the age-elevation trend discussed above. For example, the Cave des chiens is located at 142 m above the Hérault riverbed, would result in a calculated burial age for the alluvium deposits of around 1.5 Myr while the measured burial age is 3.33 $^{+0.59}/_{-0.48}$ Myr, suggesting a much lower age-elevation trend by up to a factor of ~ 2. We will discuss this discrepancy in age-elevation trends in section 4.3.

Finally, the large group of samples from Rocas, Fonctionnaire, Huttes and Leicasse caves, which are shown as grey filled data symbols in Fig 3, present burial ages at odds with any age-elevation relationship previously presented. Indeed, some caves, while being amongst the highest relatively to the river level (+ 310 m for the Rocas, + 278 m for the Fonctionnaire) present unexpected young burial ages (1.04 $^{+0.16}/_{-0.18}$ and 0.37 ± 0.18 Myr respectively). Furthermore, multiple samples from the exact same deposit of the Leicassse cave provide a very large age discrepancy (1.46 $^{+1.03}/_{-0.8}$ to 4.15 $^{+0.78}/_{-0.51}$ Myr). Theses unexpected results are discussed in section 4.4.

## 4.2 Multiple samples in the same cave level

Depending on the cave, some individual burial ages are spread over a wide age range (e.g. Leicasse, Fig. 3). Being sampled in the same deposit, such cases are an indicator of a more complex history than the classical erosion-transport-burial model assumes. One explanation is that, the wider the range, the more likely the sediments laid partially buried in a sub-aerial alluvium surface layer before being later buried in the cave when the sediments were reworked. When we compute independent burial ages, we assume that the catchment wide mean erosion rate of the paleo basin was sufficiently high to avoid significant variability in the initial $^{26}Al/^{10}Be$ (inheritance) ratio. The use of Balco and Rovey (2008) isochron method on these 3 sets of samples can be used to test this assumption. Without surprise the samples with a limited dispersion in the individual burial age computations (Escoutet and Scorpions caves) show well constrained linear regressions of $^{26}Al$ vs. $^{10}Be$ concentrations with $R^2 > 0.91$. On the other hand, samples from the Leicasse cave have a poor constraint with a regression coefficient of $R^2 < 0.2$. This is also consistent with the very large estimates of paleo-denudation rates for 2 of the 5 sampled

cobbles in the Leicasse cave, which give inconsistent values compared to all other obtained paleo-denudation rates (<50 m/Ma, Fig. 4). Omitting these two outliers, which we assume is related to a complex burial history, our results suggest no large variation in denudation rates over the last 4 Myrs (Fig. 4). We recall that, because of many uncertainties related to the paleo-denudation rate calculation (e.g. paleo-elevation of the sediment source) only the order of magnitude seems reasonable to be discussed. Therefore, rather than using the isochron method we prefer to use the independent age estimates. This choice is also supported by our observations of alluvium layers of variable thicknesses (<15 m) on the plateaus, and we advocate that the dispersion in the individual burial ages is related to the partial burial of the samples in these alluvium layers before being drained into the cave. Some would have endured partial or full burial (i.e., located within or at the bottom of the surface alluvium layer) while others would have stayed at the surface fully exposed. For the Leicasse cave samples, the conversion of TCN concentrations into burial ages is not straightforward. In this case the younger age is a better measure, equal or older, of the true burial age (~0.8 Myrs) of the cave deposit since its $^{26}Al/^{10}Be$ initial ratio was the one least likely to have been perturbed. This sample with the younger age, was then the one located closer to the surface in the sub-aerial alluvium layer prior to its drain into the cave and a deep burial preventing $^{26}Al$ and $^{10}Be$ production. The older age (~4 Myrs) is a better measure, equal or younger, of the emplacement of the surface alluvium layer that was subsequently buried in the cave. This sample was the one located the deepest in the surface alluvium layer before it was drained into the cave. We point out that theses location in the alluvium layer are relatives, that is to say, if it seems logical that the oldest being initially the deepest and the youngest the shallowest, the absolute depth prior to the final burial, however, is unknown. A few constraints can be brought by the fact that the alluvium layer thickness is usually less than 15m according to the local geological map (Alabouvette et al., 1988) and also from our field observations, which is sufficient to result in a variety of cases where production totally ceased or was only partially reduced. All the other burial ages and associated excessively high paleo-denudation rates should be used with great caution, (Fig. 4, Leicasse samples). Because of the direct cave-entrances toward the Vis River channel, and the short distance (100s meters) between the sampling site and the cave entrances, the Scorpions and the Escoutet cave samples are less likely to be affected by the injection of previously deposited alluvium at the surface of the plateaus. The narrow dispersion of independent burial ages and paleo-denudation rates are consistent with this observation. Therefore, we suggest that in the case of burial age determination for cave alluviums, if several samples are collected, independent ages should be computed and the younger one should be retained except if complications are expected due to the presence of glaciers in the valleys, which is not the case in the study area.

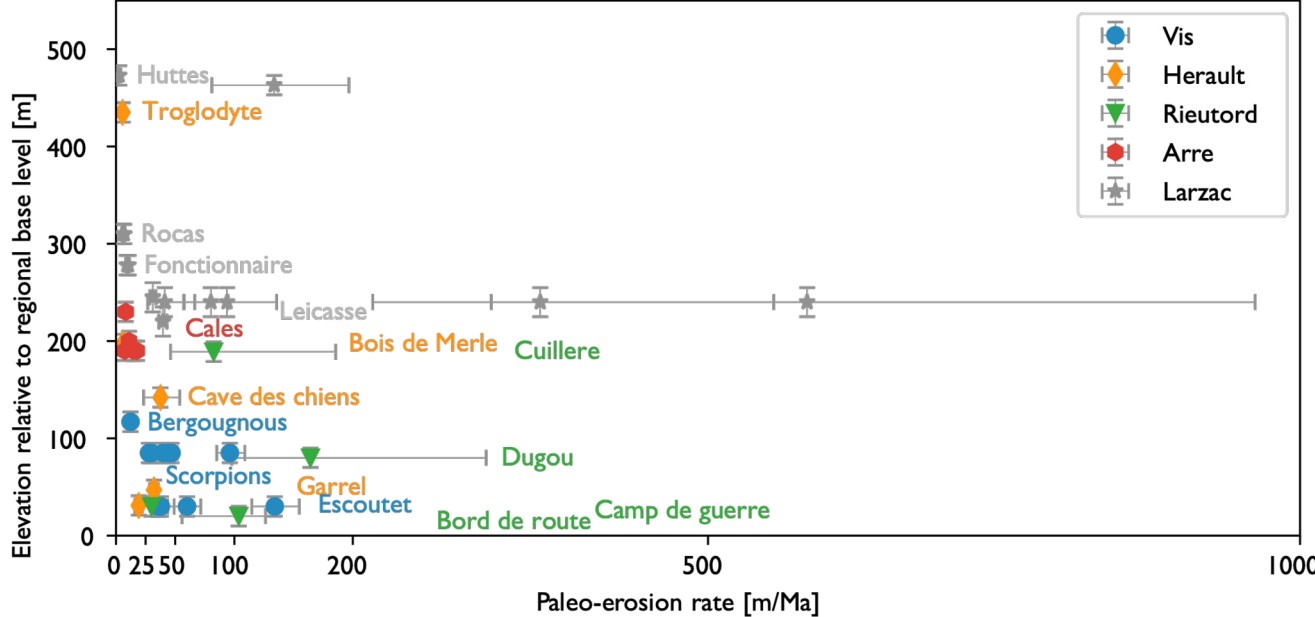

Figure 4: Paleo-erosion rate vs. relative elevation to the local water table level for each sample of this study. Caves not located on the flanks of river channel but in central plateau areas are in grey, all the other caves are adjacent to or on river flanks or canyon walls.

### 4.3 Incision rates and southern Cevennes Fault zone activity

At first glance, Figure 3 does not reveal any discernible patterns. However, when we rank the caves with respect to distance to the nearest river canyon and also their location in relation to the Cevennes Fault Zone (CFZ), three distinct sample subsets become apparent. The first one, the set of Larzac cave burial ages (Leicasse, Fonctionnaire, Rocas, Huttes; grey symbols in Fig 3) is located within the Larzac plateau (the southern plateau of the Grands Causses plateaus). All these caves are distant by 2.5 to 5 km from the nearest river channel and show no clear relationship between burial ages and relative elevations to the base level. We will discuss these caves in part 4.4. With Larzac caves set aside, all other cave burial ages (non-grey symbols) in Fig 3 have cave entrances located in river channels when base river level was close to the cave entrance elevation. These caves located within the steep flanks of river channels show a clear linear correlation - the higher the sample is above today's riverbed, the older its burial age (Fig. 3). The CFZ is a major geologic and topographic feature of the area. A first set of caves (Garrel, Bois de Merle, Cave des chiens) resides south-east of the CFZ in a lower elevation limestone plateau ~300 m a.s.l. (shown with black filled symbols in Fig 3) whilst the second includes all the caves located in the Arre, Jonte, Rieutord, Tarn and Vis river valleys and also the Troglodyte cave (white filled symbols in Fig.3) and lie north-west of the CFZ in higher elevation plateaus (600-1000 m a.s.l.). Using cave location relative to the CFZ to define our two populations we obtain an incision rate of 43 ± 6 (1σ, n=4) m/Ma for those south-east, while all other samples to the

north-west of the CFZ lead to an incision rate of 88 ± 5 m/Ma (1σ, n=32) (see Fig. 3) consistent with the local previous estimates for the Jonte valley (Sartégou et al., 2018), and for the Rieutord samples (Malcles et al., 2020a). The low differential incision rate between the two populations of ~40 m/Ma, if focused on the CFZ, could lead to earthquakes with long recurrence times, consistent with the unexpected 2019 Mw 4.9 Teil earthquake (Ritz et al., 2020). Indeed, if we consider that the 40 m/Ma is the expression of a differential uplift rate, localized on the CFZ, then the fault slip rate would be ~ 0.04 mm/yr. Using the classical relationships of Wells and Coppersmith (1994), this slip rate is expected to promote a $M_w$ 6.5 earthquake with a ~ 10 kyr recurrence time, though a longer recurrence time or lower magnitude could be an equally plausible inference, for example by taking a distributed slip rate along several faults of the CFZ. Further discussion of this observation is beyond the scope of this article and a dedicated study focused on the CFZ activity should be conducted before drawing any robust conclusion. We point out, however, that this ~ 40 m/My of uplift differential is consistent with numerical models showing that the flexural response of the lithosphere due to erosional unloading (Malcles et al., 2020) could explain this difference in incision rates without the need of seismic ruptures on the CFZ.

### 4.4 Speleogenesis implications: headward erosion of altered rock zones

The unexpected result of diminished burial ages shown in Figure 3 (when compared with the expected one using the regional trend of ~ 90 m Ma$^{-1}$) came from the 4 Larzac plateau caves that are distant of at least 2.5 km from any nearby river channel (Fig. 2 and grey filled symbols in Fig 3). These caves have a clear classical tiered morphology that can be seen on the KARST3D database (KARST3D, 2019). Quartz rich sediments in Rocas cave are only located from -20 m to -40 m deep below the surface (555 m – 575 m a.s.l.) while the deepest part humanly accessible of the cave is at -130 m (465 m a.s.l.) (Fig. 5). In Fonctionnaire cave, the sediments are in the lowest of the 3 levels, at -75 m below the surface (520 m a.s.l.). In the Leicasse cave, which has 16 km of mapped passages, alluvium is deposited in a ~1 km long passage at more than -140 m below the surface (440 m a.s.l.). The 5 Leicasse quartz cobbles sampled in a common layer at ~455 masl (coulée Borg deposit) have burial ages ranging from 0.8 to 4.1 Myrs. The Huttes cave is a 200 m long horizontal cave consisting of one level at 700 m a.s.l. Using the youngest burial age from each of the 4 caves as the closest age for sediment emplacement leads to burial ages inconsistent with that expected from epigenic speleogenesis paradigm (ESP) which would predict ages 2 to 4 Myrs older - or alternatively, a cave level elevation 150 to 250 m lower than recorded compared to the regional base level at the time of the deposit (Fig. 3).

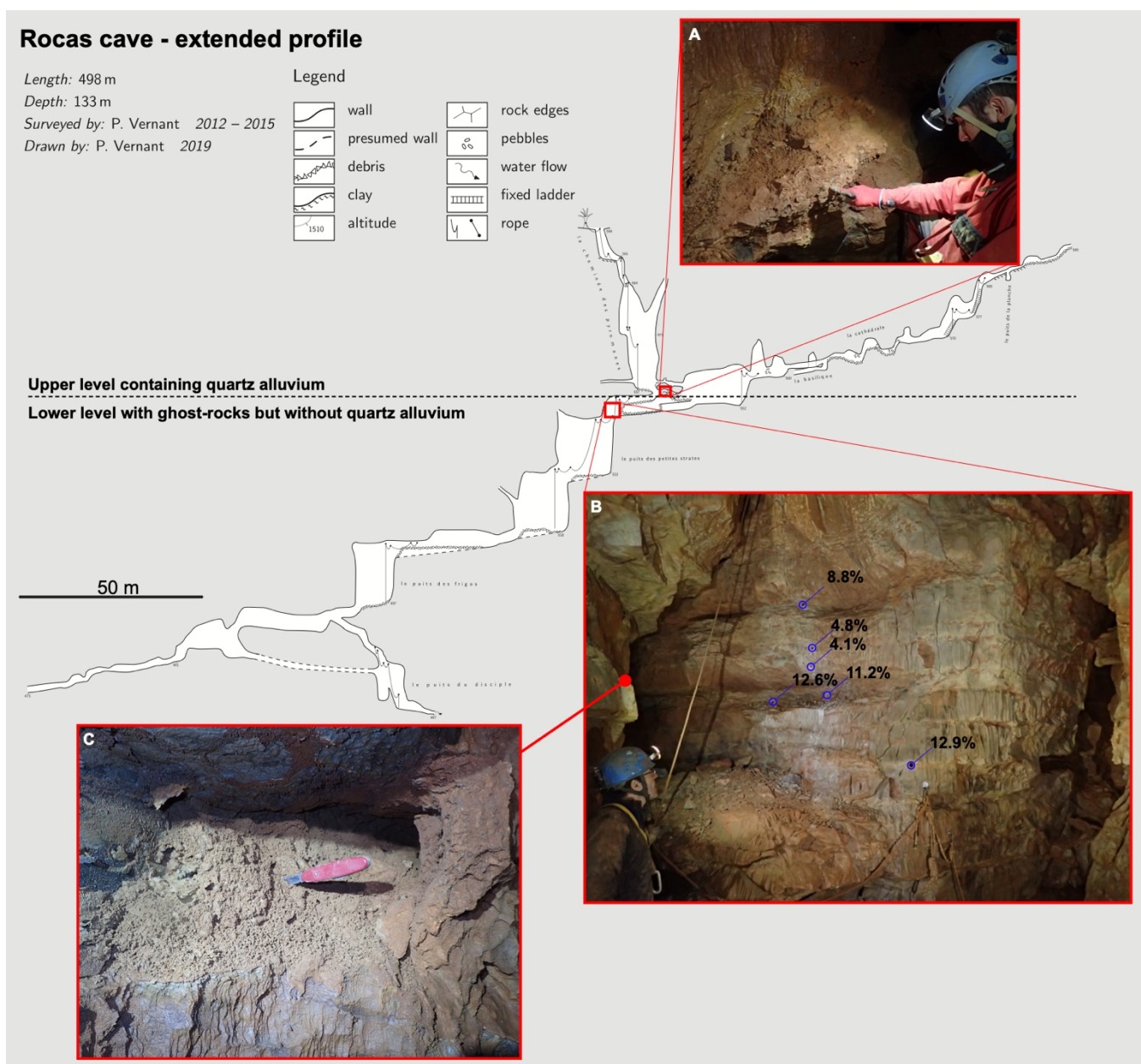

**Figure 5: Extended profile of Rocas cave, with the lower limit of the upper level containing the quartz alluvium (A) and the lower level without quartz alluvium but with ghost-rocks. Example of cave wall with higher porosities near the beading planes showing the ghost-rocks (B), sometimes the porosity is so high that the rock structure is still apparent but fully pulverulent (C).**

The absence of sediments below 40 m depth in Rocas cave indicates that the lower galleries formed less than a 1 million years ago, after emplacement of alluvium in the cave's highest level. In this younger part of the cave, passages show morphologies similar to those reported for ghost-rock caves by Dubois et al. (2014) and Rodet (2014), that is to say the cross section of the galleries are characterized by lens (or half lens) shape extending in the weathered strata while the non-

weathered strata and the lower part of the galleries are characterized by scallops, potholes and incised meanders. Furthermore, the preserved ghost-rock at the type of the lens shape have porosities larger than 10 % (Fig. 5). In the ESP model, the meteoric water moves downwards through plateau bedrock, enlarging fractures and bedding planes to form the cave gallery at the contemporary base level. In contrast, the sub horizontal upper level of the Rocas cave, which was filled with sediments 1 million years ago, is clearly not related to the base level of the Vis River, located at that time 250 m lower, as registered by the same 1 Ma burial ages for the Scorpions and Bergougnous cave samples (Fig. 3). The same rationale applies to Fonctionnaire, Leicasse and Huttes caves and no impervious layers of marls are reported in the stratigraphic log at these cave elevations that could produce a perched karst. Furthermore, the wide range of the 5 cobble burial ages for the Leicasse (Fig. 3) and paleo-denudation rates (Fig. 4) are significantly larger than the ranges for all the caves located within river channels; (e.g., the 5 cobbles for Scorpions). We conclude that the sediments emplaced in Leicasse were not transported into the cave quickly nor directly by riverine fluvial processes, but they had resided at shallow depth within sub-aerial alluvium layer at the surface of the plateau at least for a period of 3 Myrs, the difference between the youngest and the oldest measured cave burial ages of the 5 samples. Alluvium of former poljes are present at the surface of the Larzac plateau and could be the source of the sediments that were later buried in the cave at the -140 m depth level without direct relation with the contemporary base level.

Based on these results, we propose that contrary to the ESP, the speleogenesis of the Larzac plateau is driven by karstic headward erosion from the canyon walls to the center of the plateau rather than by water working its way from the top of the plateau toward the valley. Given the rather quick formation of the caves, we propose that the passages were pre-structured by an alteration phase under low hydrodynamic gradient leading to numerous incipient passages full of ghost-rocks or isovolumic alterite, as schematically described in Figure 6, retaining the original rock structure, and called primokarst (Rodet, 2014). Ghost-rocks remain trapped in incipient passages where the water flow is practically absent since the boundaries of these incipient openings are impervious rocks, very thin fissures or bedding planes allowing only water and ions to slowly flow through. When the canyon cuts through one of these passages, it opens an outlet large enough to create a high hydrodynamic gradient allowing the mechanical removal of the ghost-rocks and creation of a new cave in a fairly short time as experienced in real time in Belgian quarries (Quinif, 2010). This is what occurred after 1 Ma for the lower part of the Rocas cave and around 0.3 to 0.5 million years ago, for the Fonctionnaire cave (Fig. 6). This headward erosion works its way from the canyon walls toward the center of the plateau following the primokarst structures, and possibly creates deep sump (>100m) rather than a river related tiered cave. Once the voids are opened, the water can flow through quickly and modify the cave morphology, enlarging it and creating hydrodynamic markers like scallops. The specificity of the Larzac plateau with sparse and thin deposits of quartz rich alluvium across its surface, has led to these unexpected results showing that cave levels, at least in this region, are related to preferential alteration levels that are subsequently emptied and, in some cases, enlarged by underground rivers when the primokarst was near or below the base level. While previous authors have already proposed that ghost-rock removal could lead to large networks (Dubois et al., 2014, Quinif and Bruxelles, 2011), our results show that this process can be the major mechanism in the speleogenesis of large limestone plateaus like the Larzac

(1000km²). These new observations also suggest that karstification can be a continuum process starting with hypogenic/ghost-rock karstification and continuing with epigenic processes. The main difference of ghost-like karstification with the widely accepted ESP model is that the network cave geometry is already established during a hypogene/ghost-rock phase and that evolution of the base level with its associated water gradient modification is a subsequent phase mostly responsible for the opening of the voids with little control on their structure. As pointed out by Dubois et al. (2022) karst morphologies are used by scientists to speculate on processes that induce speleogenesis. It leads to a tremendous number of different processes to form caves (see for example Figure 3 of Harmand et al., 2017). Here we choose to follow an approach driven by the principle of parsimony also termed as Ockham's razor and propose a continuum process where cave geometry complexity is only driven by the primary phase of alteration. We are not the first authors to do so (e.g. Dubois et al, 2014, 2022), but acknowledge that this is an ongoing debate as attested by the discussions with the reviewers triggered by the first drafts of this study. Quinif (2010) suggested the need for a new paradigm about karstogenesis implying ghost-rock processes. We know from the history of sciences that shifting from one paradigm to another is a complex journey (Kuhn, 1962). More studies and debates will be needed to overcome the present matter of contention about how ghost-rock processes should be considered in karstogenesis, that is to say, rather as a secondary process (e.g., Schmidt, 1974, Klimchouk, 2012) or the primary process (e.g., Rodet, 2014, Dubois et al., 2014 and this present study).

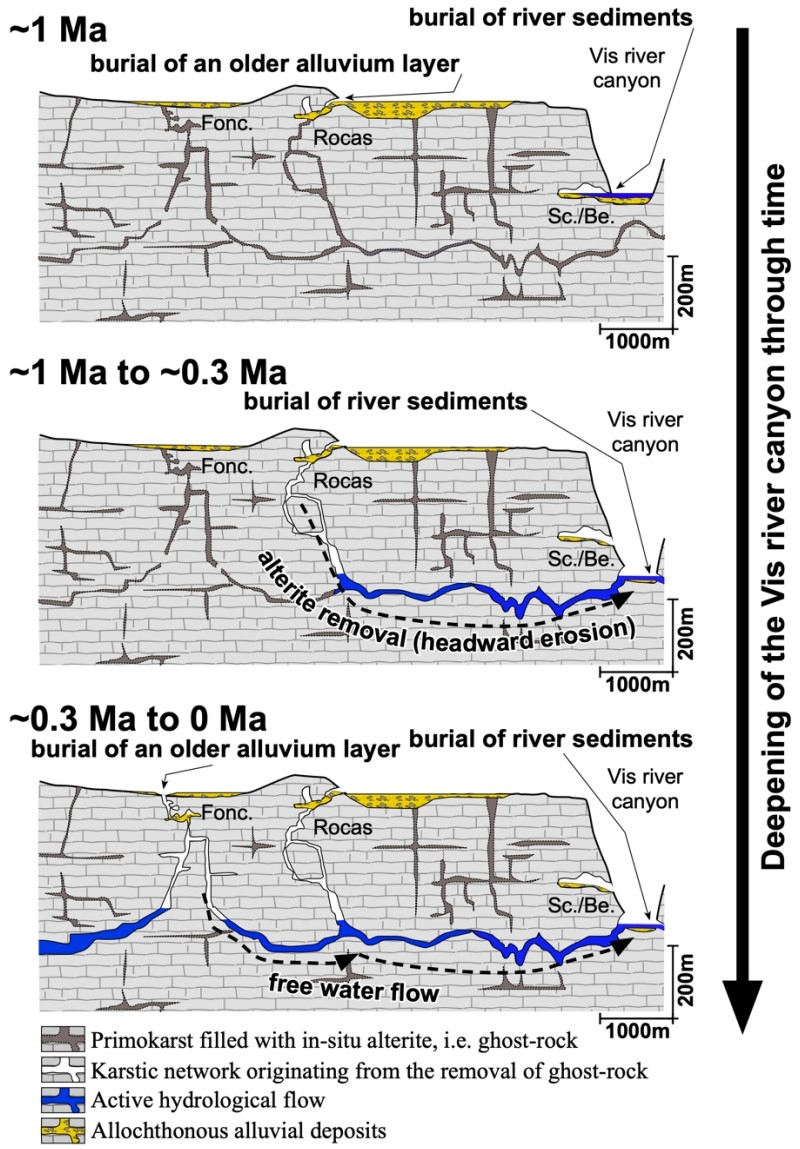

**Figure 6: Proposed Speleogenesis model for the cave in the center of the Larzac plateau based on the burial ages obtained in this study. The speleogenesis of the area is primarily due to alterite (ghost-rock) removal due to underground headward erosion. Fonc.: Fonctionnaire cave, Sc./Be.:Scorpions and Bergougnous caves.**

## 5 Conclusions

Combining 22 new burial ages with 15 previously published ones, we propose a mean regional river incision rate of 88 ± 5 m/Ma for the Grands Causses region and the first incision rate for the Herault river of 43 ± 6 m/Ma, both over the last ~4

Myrs. These two regions are separated by the Cevennes Fault Zone, which could accommodate part of this ~ 40 m/My of differential uplift as suggested by the Mw 4.9 Teil earthquake surface rupture ~100 km afar to the north east. The use of thirteen more new burial ages for quartz-rich alluvium deposited in the 4 caves located in the central Lazarc plateau region which are all located about 2-3 km distant, detached from the other caves we sampled on the flanks of river channels and thus incision could not have been supplied by fluvially transported riverine sediments, give unexpectedly younger ages by 1-

3 Myrs from that predicted by the conventional epigenic speleogenesis model. We conclude that the speleogenesis in the study area does not follow the widely accepted epigenic paradigm but is primarily due to headward erosion of previously altered rocks. Once the river cuts through a primokarst by deepening its canyon the induced high waterflow can evacuate the ghost-rocks and quickly form new caves. Some of these caves can show several levels whose timing of construction are only related to the time of alteration of the rocks prior to the speleogenesis rather than being correlated to regional base level

changes. We suggest that the previously proposed ghost-rocks process for large karst network genesis (Dubois et al., 2014, Quinif and Bruxelles, 2011) can also be applied at the scale of large limestone plateaus and could be the first stage of large void opening prior to the high waterflow hydrodynamic phase.

**Competing interests**

The contact author has declared that none of the authors has any competing interests.

**Acknowledgments**

We thank C. Mifsud, K. Stevens and S. Kotevski for their assistance during the sample processing. We thank K. Wilcken for assistance in some of the earlier AMS measurements. We also thanks Y. Guessard, L. Bruxelles, M. Roux and L. Leterme for the identification of cave infilling or their help during sampling. We acknowledge financial support from ANSTO portal AP12168 and the financial support from the Australian Government for the Centre for

Accelerator Science at ANSTO through the National Collaborative Research Infrastructure Strategy (NCRIS).

**Financial support**

O. Malcles benefited from a PhD grant provided by the French Ministère de l'Éducation Nationale de l'Enseignement Supérieur et de la Recherche and an AINSE-ANSTO French Embassy Research Internship (SAAFE) portal This research was partially supported by ANSTO portal project AP12168.

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
