# Peer review of "Cosmogenic nuclide-derived downcutting rates of canyons within large limestone plateaus of southern Massif Central (France) reveal a different regional speleogenesis of karst networks"

_EGUsphere, 2023_

## Author Response (AR1)

**Answer to R1 (Fritz Schlunegger):**

I enjoyed reading the paper (despite the hurdles listed below; these are not scientific but editorial) because the topic is interesting, and the data including the story and conclusions should be published. I thus congratulate the authors to have taken such an endevour.

***Thank you. It was quite an endeavour as you have guessed it, but it was a pleasant one and we hope that the data will be helpful to the communities interested in the karst dynamic or the intraplate deformation.***

However, the paper needs a substantial editorial improvement. This includes the English, which needs to be polished and carefully checked for typos/grammars (I am not native speaking, but I detected many sentences where the grammar is not correct).

***English grammar and typos will be carefully checked. We'd like to point out that one of the authors is Australian (native speaker) and that three others have spent several years in Australia or in the USA.***

The figures need to be improved as well they are hard to read, and sometimes it is not obvious how they have to be read. For instance, in Figure 3, I see abbrevations such as Vis, Herault etc. on the right side. Are these valleys or regions? Then I see labels such as Cales, Deves-Mejean etc. in the main part of the figure. Are these caves? I though so, but it is not well explained.

***Vis, Hérault, etc. are indeed valley. Other cited labels (e.g. Cales) are indeed caves. We'll state it carefully not only in the main text but also in figure captions to avoid any misunderstanding.***

I also thought that the photo of Figure 2 is not clear. I don't really see the Vis valley. The sunlight appears to perturb the impression of a dissection.

***We will replace the picture with a better one showing more clearly the Vis canyon.***

In the same sense, section 4 is not well organized. It starts with a discussion, but the data has not been presented yet - a results section is missing. I do see that the results are all presented in the discussion section, but this needs to be clearly separated.

***We will do our best to present the results prior to the discussion to avoid confusion.***

Finally, I understand that the Leicasse cave system records a different story than all the other ones, and the major arguments are put forward in section 4.3. This part, however, needs to be better and more carefully explained. For instance, I could not really follow the arguments about recycling of the quartz-rich sediments, but perhaps I missed the point here.

***Not only the Leicasse cave deposits are recording a different story but the Fonctionnaire and the Rocas ones as well. However, the main argument supporting the ghost-rock dynamic is indeed brought by the Leicasse cave. Explanation in the main text will be extended for clarity. We will also add a figure to illustrate the recycling of the sediment.***

One question I have not understood: Why did the burial of quartz pebbles stopped at 1 Ma?

***The burial of quartz pebbles is an ongoing process and did not stopped at 1 Ma. For the cave entrance located in the valley, the sampling is biased by the known caves. For the "plateau" caves, the apparent burial age is controlled by the timing of the cave opening by regressive***

*erosion. We don't have enough data to draw robust conclusions, but the farther from the canyon the cave is, the younger the burial age (this is based only on 3 caves so we won't mention it in the text, we first need to find more cave with sediments at different distances from the canyons).*

My review might appear as critical, but nevertheless I really enjoyed this paper because it is a nice contribution, but there is space for making it more reader-friendly.

*Thank you again, we appreciated your comments and suggestions that will help us to substantially improve our manuscript.*

Please also cite Haeuselmann et al. (2009, Geology) as this is a classical paper addressiing the same topic.

*We did not find Häuselmann et al. (2009, Geology) in Philipp's publication list: [http://www.sghbern.ch/praezis/publist.pdf](http://www.sghbern.ch/praezis/publist.pdf), so we will refer to Häuselmann et al. (2007, Geology) paper since it is a classical paper on this topic.*

**Answers to R2 (Philippe Audra):**

Overall, the article is well-written, the discussions are based on the results, and are supported by rather high-quality figures.

Nevertheless, some points, of varying importance, remain to be clarified or reformulated.

***Thank you, here is how we propose these issues to provide an improved manuscript.***

Minor corrections :

- 40: *...advocates of the conventional model postulate that...* => references ?

***Excellent question! It appears that this removal of insoluble particles is implicitly assumed in the literature. Without going into details of the epigenic speleogenesis paradigm (impact of limestone purity, proposed thresholds for hydrodynamic behaviour linked with fracture size, etc. we will refer to Dubois et al. (2014) and Quinif (2010) who discuss the idea of soluble and insoluble products removal.***

- 58-60: *while hypogenic and ghost-rock karstification occur below the base level and subsequent tiered karst geometries cannot be interpreted in terms of river entrenchment phases* => references? This is not entirely true, or at least not well formulated. Fantomization initially acts below the base level, but the second phase of removal is dependent on the base level, enabling solid particles to be washed towards the outlets. For hypogenic, the upward flow and corresponding caves are also below base level by definition, but the outlet and connecting conduits may present horizontal extensions connected to the base level, and thus considered as records of the progressive incision of the hydrographic network.

***We agree that the second phase is dependent on the base level but only incidentally, when the surface cut through the primokarst. Once the primokarst has been emptied from its alterite, the geometry of the karst network will follow the primokarst geometry with secondary modifications of the network according to the hydrodynamic regime and evolution depending on local conditions. Clearly, our main point is that the overall geometry of the karstic network is controlled by the primokarst and not the base level. In other word, the location of a horizontal level is expected to be linked with the location of a horizontal ghost-rock area and not by a hypothetical stability of the base level. The location of the horizontal ghost-rock area itself being induced by any heterogeneity promoting alteration due to slightly higher permeability. As mentioned by other authors (e.g. Dandurand et al., 2019), fantomization can be induced by convection cells and later on solid particles can be washed away even if they are below the base level, as long as the outlet is lower than the entrances and that potential energy is sufficient to move the particles. In this case, as mentioned by Klimchouck (2017), fantomization could be part of hypogene processes, or we might wonder if hypogene processes could be part of Fantomization! This debate is beyond the scope of this manuscript, but we will modify the text to describe Dandurand et al. (2019) ideas in the introduction to refer to it later in the discussion. As a matter of fact, we are afraid that we might agree to disagree with Philippe Audra on the fact that horizontal extensions might be representative of the progressive incision of the hydrographic network, since it is one of the main results of this study, at least for our study area, and likely for some other areas as proposed by Dandurand et al. (2019). We would like to remind that we are***

*not pleading for a paradigm change where all the horizontal conduits formation could not be linked to base level evolution, but we show that in our study area it is not likely the case.*

- Fig1: the allochthonous alluvial deposits give the impression that they are injected from the valley into the caves, which is possible but limited to the entrance areas. I assume that the sediments are transported from the plateau and stored in the horizontal level => 2 figures distinguishing fluvial alluvium and subterranean sediments would be required.

*We agree that in the general case, sediments can be flushed down the cave if rivers flow from crystalline mountains toward limestone plateaus. To add clarity about sediments recycling as requested by reviewer #1 we will add a figure which will also address this comment. In our case, the sediments that can be used to estimate the river downcutting rate have to come directly from the river. This is supported by the fact that cave deposits are usually near the entrance and that the cave geometries (Scorpions, Escoutet) are probably associated with endokarstic loops. We choose this representation for the sake of clarity given the fact that the precise source of the sediment will not change the main conclusion of our study. However, we will add a comment, at least, in the figure caption to make it clear that two sources of quartz rich sediments can be found in the area.*

- 155 : *Reported layer thicknesses are usually less than 15m =>* reference?

*We will modify the text to refer to the geological map of the area and to our field observations where we have never seen layers thicker than a few meters.*

- 180: *The low differential incision rate between the two populations of ~40m/Ma, if focused on the CFZ, could lead to earthquakes with long recurrence times, consistent with the unexpected 2019 Mw 4.9 Teil earthquake (Ritz et al., 2020) =>* better explain the relationship between the low dispersion of incision rates along the CFZ and low occurrence seismicity, it's not clear at all. Similarly, this point in the conclusion needs to be clarified.

*We will expand this discussion in the revised manuscript based on the following rational: if we convert the 40m/Ma incision difference in differential uplift (supported by low river slope gradients of the area), and if this is all accommodated by the CFZ (which has yet to be proven, it could be distributed over a 10-120 km width band long the CFZ), the fault slip rate would be 0.04 mm/yr. According to Wells and Coppersmith (1994), this slip rate would produce for example magnitude ~6.5 earthquake earthquakes with recurrence time of ~10 000 yrs, if off fault or distributed deformation occurs, the recurrence time would greatly increase. We will also point out that it is consistent with numerical models showing that the flexural response of the lithosphere due to erosional unloading (Malcles et al., 2020) could explain this difference in incision rates without needing the CFZ to slip.*

- Fig4: the legend mentions "*unfilled data points*" which are not visible on the graph.

*Unfilled modified to "white filled" for consistency with the "black filled" ones.*

- Primokarst: make it clear in the introduction what this term means in relation to ghost weathering, and ensure that their alternative use actually corresponds to a difference, if any. Otherwise, just use the term ghost.

*A brief description of the primokarst will be added in the introduction. We consider that the ghost weathering is more tightly linked with the processes itself of isovolumetric alteration, and*

*therefore can be local or isolated, while the primokarst refers to the incipient karst network geometry not yet created.*

- Fig6: reword legend: 2. Karstic network originating from the focused removal of ghost

*We will reword the legend as requested.*

- 260: *We suggest that the already proposed ghost-rocks process for large karst networks genesis (Dubois et al., 2014, Quinif and Bruxelles, 2011) can also be applied at the scale of large limestone plateaus and could be the first stage of large void opening prior to the high waterflow hydrodynamic phase =>* by definition, ghost karstification doesn't create "large voids", it prepares them: reformulate

*We agree, and this is what we meant by "the first stage". We will reformulate accordingly to make more it evident.*

Major corrections :

- Fig5a: a much more precise description of the sedimentary profile is needed, rather than a photograph where nothing in particular can be seen, with the caption mentioning levels with or without quartz. What is the nature of this sediment? Fluvial, debris-flow? More generally, all sampled sites should at least be described (profile of the cave with location of the sample as shown in Fig. 5, detailed description of the profile of the section studied, nature of the dated elements (quartz pebbles or amalgam of sands?). This aspect is crucial for assessing the representativeness of the age and the type of interpretation that can be made of it. Generally speaking, this lack is the article's main negative point. Descriptions do not necessarily need to be included in the text and can be referred to in supplements.

*We will provide in a supplementary file the sample locations on the cave maps, the link to the 3D cave models if available on the Karst3D data base ([https://data.oreme.org/observation/karst3d](https://data.oreme.org/observation/karst3d)). All the sites are related to fluvial deposits, but we will also add a brief description of the outcrops where the samples were collected, as well as pictures of the sites. In the already provided supplemental table we will add the nature of the samples (i.e., single cobble of gravel amalgam).*

- Similarly, it is essential to provide all geochemical dating data in a table accessible in the supplements. No article should be accepted without the provision of objective data, which can also be later used to critically analyze the results.

*The relevant data were already provided in the supplementary table (https://egusphere.copernicus.org/preprints/2023/egusphere-2023-765/egusphere-2023-765-supplement.pdf). The nature of the sediment dated (i.e., single cobble, or gravel amalgam) will be added (see also answer to the previous comment).*

- 240*: The main difference with the widely admitted ESP model being that the network geometry is defined by the hypogene/ghost-rock phase and not by the base level time evolution*. I disagree with this conclusion, which ties in with my comment on lines 58-60. The fantomization phase determines the presence of discontinuities (below the base level), which will later be used for mechanical removal of the fantoms along the karst flow axes, but this occurs 1/ according to the position of the base level (not below it), and 2/ as the authors suggest, regressively from the valleys towards the center of the plateau, with the local establishment of horizontal conduit levels clearly correlated with the base level (the blue conduits on Fig. 5 show this very well).

*One again we will have to agree to disagree on point 1, as already proposed by Dandurand et al. (2019), removal of ghost rocks can occur below the base level and form sumps or long conduits deeper than 100m below the base level. And even though we would agree that ghost rock was removed only at the base level and above, the horizontal segment of the conduits would have nothing to do with the base level elevation, hence they could not be used as a marker of the base level steadiness. As mentioned in the answer to comment on lines 58-60, we will expand the introduction to describe Dandurand et al. (2019) ideas about drowned conduits formation related to removal of ghostrocks deep (i.e., >100m) below the base level.*

- 230: *This regressive erosion works its way from the canyon walls toward the center of the plateau following the primokarst structures, and possibly creating deep sump (>100m) rather than river related tiered cave*. This assertion about the origin of drowned conduits at great depth should mention the sites concerned (source of the Vis? Others?), and must clearly state that it is a suggestion, unverified at present, incidental to the subject of this article.

*One again we will have to agree to disagree with Philippe Audra, many drowned conduits are known and dived at more that 100m below the base level in our study area (e.g. Gourneyras spring with explored drowned galleries down to 100 m, while the network shows mostly a horizontal geometry ([https://www.plongeesout.com/sites/roussilon-pyrenees/herault/Gourneyras.htm](https://www.plongeesout.com/sites/roussilon-pyrenees/herault/Gourneyras.htm)), Vis spring, Durzon spring). These deep phreatic conduits are hardly explained by any other processes. We will expand the description of the context to describe these drowned conduits together with Dandurand et al. (2019) ideas about other drowned conduits explained by the ghost-rock pre-structuration as the Fontaine de Vaucluse or the Touvre spring that are not in our study area. Given that it has already been proposed for other regions, and that it is likely a substantial process in the speleogenesis in our study area, we think that it is not incidental to the subject. To contrary, it is closely related to the subject and ask the question: is the paradigm of conduit development related to base level steadiness verified at the present? We are not willing to push as far since a lot of work is required before drawing any conclusion, but we are not the first ones to state it and we strongly believe that our study brings elements of discussion to this ongoing debate.*

---

## Referee Report (RR1)

The paper of Malcles et al. presents burial ages obtained in karstic networks of southern Massif Central. The authors propose for networks far from the river valley flanks or cliff walls that the well accepted epigenic speleogenesis model (network is formed when water table is stable then abandoned when the river is lower due to incision) cannot be applied and propose a model based on speleogenesis controlled by regressive denudation towards inner part of the plateau.

Despite I am a bit far from this topic but more attracted by the cosmogenic nuclide applications, I think that these data have to be published after rewriting with more explanations and simplification. At this stage, some parts of the paper are a bit fuzzy, and the cosmogenic methodology lacks important information. See pdf

Part 3 :

      Please provide the types of spikes used and their concentrations.
      Precise the spallation production rate used.
      What half-lives have been used for 10Be and $^{26}$Al?
      What is the spallation production rate ratio used for $^{26}$Al/$^{10}$Be (6.75?)

Line 109 – 123: the age calculation explanations are not clear and difficult to understand.
      Using the data set provided I have recalculated all ages and paleo denudation rates (see excel table at the end of this review) using a normal approach sample by sample, ignoring postproduction. The clauside amalgam can be modelled (2.04±0.46 Ma and 147.8 ± 33.16 m/Ma).

      A banana plot will help to have in one figure the entire dataset.

[Figure]

Regarding the production rate used in the calculation we do not know if it the one of the cave location or the one of the sources of the sediment (mean production rate of the watershed). This will not alter the burial ages but will highly influence the paleo denudation rate determination.

Fig.3. try to use different symbols for a given site. This will help the reader working on black and white paper sheet. In this figure you have plotted two Rocas ages and two Fonctionnaire ages corresponding to two measurements on the same samples. If this is true do not present both data as this will give artificial more weigh to these ages. You can do this when working on different samples.

Line 140-144: the use of isochron approach is not helpful here.

Line 145-162: This par is hard to understand!! You are explaining that samples might have been already buried prior to they are deposited in the network; this yields to a scattering in the age distribution. How can you know the sample position in the alluvium cover before its burial in the network? (Line 52-155:" *This sample with the younger age, was the one located closer to the surface in the surface deposited alluvium layer prior to burial. The older age (~4 Myrs) is a better measure, equal or younger, of the emplacement of the alluvium layer that was subsequently buried into the cave. This sample was the one located deeper in the surface alluvium layer before cave burial*")

Line 179-180: What is the mean displacement rate of the CFZ fault, and the mean offset after earthquakes? In Ritz et al. one can find max offset values of 20 cm and it is also mentioned in the same paper that no surface deformation was observed during historical seism.
        Can you thus conclude that this fault can be responsible of the incision of the studied valleys? What about a global uplift due to Massif Central Mountains?

Fig. 4; change symbols and change police type for network far from the river cliffs.

Line 188: What do you mean by "**The unexpected result of diminished burial ages shown in Figure 3…**"?

Line 198. Can you explain you approach here:" *speleogenesis paradigm (ESP) which would predict ages 2 to 4 ma older - or alternatively, a cave level elevation 150 to 250m lower than recorded compared to the regional base level at the time of the deposit)?*

Line 202: Why the absence of sediment in Rocas implies an age younger than 1Ma?
Line 208: Scorpions and Bergougnous sites seem to be affected by the Vis River. Why do you compare the Rocas sediments (from alluvial deposits on top of the plateau) with these two sites?
Why the same age of 1Ma cannot be related to the activity of the entire network from Sc/Be to Rocas?
As you proposed a new formation model it is worth better explaining this last part synthetized by fig. 6 and show how you construct the chronology from 1Ma to present.

| | Long | Lat | Alt | Pres. (mbar) | Stone sc | Lal sc | Be 1e4at/g | ± 1e4at/g | Al 1e4at/g | ± 1e4at/g | Paleo denudation m/Ma | Age Ma | ± Ma | R modeled | R measured | 10Be model 1e4at/g | 26Al Model 1e4at/g |
|---|---|---|---|---|---|---|---|---|---|---|---|---|---|---|---|---|---|
| Bergougous | 3.493 | 43.889 | 426 | 963.11 | 1.43 | 1.43 | 20.15 | 0.4 | 77.87 | 3.46 | 11.38 ± 0.61 | 1.16 | 0.06 | 4.04 | 3.86 | 20.00 | 80.72 |
| Scorpions | 3.625 | 43.885 | 275 | 980.65 | 1.25 | 1.25 | 5.73 | 0.15 | 25.25 | 1.12 | 42.27 ± 2.36 | 0.95 | 0.05 | 4.56 | 4.41 | 5.68 | 25.90 |
| Scorpions | 3.625 | 43.885 | 275 | 980.65 | 1.25 | 1.25 | 6.04 | 0.13 | 26.4 | 1.09 | 37.85 ± 1.94 | 1.04 | 0.05 | 4.37 | 4.37 | 6.04 | 26.40 |
| Scorpions | 3.625 | 43.885 | 275 | 980.65 | 1.25 | 1.25 | 9.63 | 0.24 | 43.63 | 1.83 | 27.04 ± 1.44 | 0.80 | 0.04 | 4.88 | 4.53 | 9.43 | 46.04 |
| Scorpions | 3.625 | 43.885 | 275 | 980.65 | 1.25 | 1.25 | 2.53 | 0.08 | 9.82 | 0.58 | 81.13 ± 5.71 | 1.30 | 0.09 | 3.90 | 3.88 | 2.53 | 9.85 |
| Scorpions | 3.625 | 43.885 | 275 | 980.65 | 1.25 | 1.25 | 8.15 | 0.2 | 33.57 | 1.35 | 26.06 ± 1.35 | 1.16 | 0.06 | 4.12 | 4.12 | 8.15 | 33.57 |
| Scorpions | 3.625 | 43.885 | 275 | 980.65 | 1.25 | 1.25 | 6.17 | 0.17 | 25.15 | 1.22 | 34.54 ± 2.07 | 1.18 | 0.07 | 4.10 | 4.08 | 6.16 | 25.25 |
| Escoutet | 3.626 | 43.885 | 220 | 987.10 | 1.19 | 1.19 | 5.72 | 0.13 | 20.44 | 1.28 | 30.51 ± 2.14 | 1.48 | 0.10 | 3.56 | 3.57 | 5.72 | 20.36 |
| Escoutet | 3.626 | 43.885 | 220 | 987.10 | 1.19 | 1.19 | 6.32 | 0.15 | 26.32 | 2.37 | 36.49 ± 3.49 | 0.95 | 0.09 | 4.56 | 4.16 | 6.28 | 28.65 |
| Escoutet | 3.626 | 43.885 | 220 | 987.10 | 1.19 | 1.19 | 2.16 | 0.11 | 10.08 | 0.51 | 123.97 ± 9.3 | 0.77 | 0.06 | 5.01 | 4.67 | 2.08 | 10.42 |
| Escoutet | 3.626 | 43.885 | 220 | 987.10 | 1.19 | 1.19 | 4.47 | 0.12 | 20.4 | 2.72 | 66.79 ± 9.2 | 0.48 | 0.07 | 5.71 | 4.56 | 4.42 | 25.23 |
| Troglodyte | 3.469 | 43.744 | 485 | 956.33 | 1.50 | 1.50 | 5.17 | 0.13 | 2.47 | 0.63 | 4.54 ± 1.17 | 5.64 | 1.45 | 0.48 | 0.48 | 5.17 | 2.47 |
| Cave | 3.619 | 43.759 | 218 | 987.33 | 1.19 | 1.18 | 2.18 | 0.06 | 3.61 | 0.91 | 35.39 ± 9.01 | 3.12 | 0.79 | 1.66 | 1.66 | 2.18 | 3.61 |
| Bois | 3.614 | 43.755 | 273 | 980.88 | 1.25 | 1.25 | 8.44 | 0.2 | 9.96 | 1.72 | 6.22 ± 1.09 | 3.74 | 0.66 | 1.18 | 1.18 | 8.44 | 9.96 |
| Garrel | 3.615 | 43.834 | 184 | 991.34 | 1.15 | 1.15 | 9.26 | 0.21 | 30.51 | 3.47 | 20.62 ± 2.43 | 1.22 | 0.14 | 3.99 | 3.29 | 9.17 | 36.59 |
| Garrel | 3.615 | 43.834 | 200 | 989.45 | 1.17 | 1.17 | 7.26 | 0.17 | 30.32 | 2.16 | 28.75 ± 2.24 | 1.09 | 0.08 | 4.27 | 4.18 | 7.24 | 30.91 |
| bord de route | 3.709 | 43.954 | 175 | 992.40 | 1.14 | 1.14 | 3.54 | 1.18 | 21.6 | 0.15 | 119.5 ± 39.93 | 0.02 | 0.01 | 7.11 | 6.10 | 3.04 | 21.60 |
| Camp | 3.71 | 43.955 | 190 | 990.63 | 1.16 | 1.16 | 8.87 | 3.28 | 42.9 | 0.31 | 59.41 ± 22.01 | 0.02 | 0.01 | 7.06 | 4.84 | 6.08 | 42.90 |
| Dugou | 3.71 | 43.957 | 245 | 984.16 | 1.22 | 1.22 | 1.27 | 0.33 | 4.29 | 0.06 | 136.22 ± 35.57 | 1.61 | 0.42 | 3.38 | 3.38 | 1.27 | 4.29 |
| Cuillere | 3.71 | 43.957 | 354 | 971.44 | 1.34 | 1.34 | 1.7 | 0.53 | 3.75 | 0.07 | 794.48 ± 248.73 | 0.03 | 0.01 | 7.11 | 2.21 | 0.53 | 3.75 |
| Clauside | 3.71 | 43.96 | 487 | 956.10 | 1.51 | 1.51 | 1.14 | 0.08 | 3.16 | 0.67 | 147.8 ± 33.16 | 2.04 | 0.46 | 2.77 | 2.77 | 1.14 | 3.16 |
| balcony) | 3.537 | 43.964 | 470 | 958.05 | 1.49 | 1.49 | 24.73 | 0.57 | 64.52 | 2.79 | 5.83 ± 0.31 | 2.03 | 0.11 | 2.61 | 2.61 | 24.73 | 64.52 |
| Gr. | 3.537 | 43.964 | 470 | 958.05 | 1.49 | 1.49 | 7.67 | 0.17 | 13.41 | 1.67 | 16.17 ± 2.07 | 2.51 | 0.32 | 2.17 | 1.75 | 7.60 | 16.49 |
| Gr. | 3.537 | 43.964 | 470 | 958.05 | 1.49 | 1.49 | 7.87 | 0.18 | 16.6 | 1.12 | 15.15 ± 1.13 | 2.57 | 0.19 | 2.11 | 2.11 | 7.87 | 16.63 |
| Gr. | 3.537 | 43.964 | 480 | 956.90 | 1.50 | 1.50 | 14.67 | 0.31 | 36.02 | 2.88 | 9.26 ± 0.79 | 2.25 | 0.19 | 2.41 | 2.46 | 14.69 | 35.40 |
| Gr. Entrance | 3.537 | 43.964 | 510 | 953.46 | 1.54 | 1.54 | 20.27 | 0.5 | 50.83 | 3.07 | 7.1 ± 0.49 | 2.14 | 0.15 | 2.51 | 2.51 | 20.27 | 50.83 |
| Rocas | 3.482 | 43.841 | 310 | 976.56 | 1.29 | 1.29 | 41.35 | 0.87 | 171.96 | 7.17 | 5 ± 0.26 | 1.02 | 0.05 | 4.16 | 4.16 | 41.35 | 171.96 |
| Rocas | 3.482 | 43.841 | 310 | 976.56 | 1.29 | 1.29 | 47.34 | 1.56 | 175.4 | 6.7 | 3.75 ± 0.21 | 1.24 | 0.07 | 3.67 | 3.71 | 47.52 | 174.40 |
| Fonctionnaire | 3.467 | 43.805 | 278 | 980.30 | 1.25 | 1.25 | 39.73 | 0.78 | 207.82 | 9.87 | 7 ± 0.39 | 0.45 | 0.03 | 5.51 | 5.23 | 39.42 | 217.38 |
| Fonctionnaire | 3.467 | 43.805 | 278 | 980.30 | 1.25 | 1.25 | 38.89 | 0.98 | 226.86 | 8.79 | 8.42 ± 0.43 | 0.19 | 0.01 | 6.28 | 5.83 | 38.00 | 238.80 |
| Huttes bot | 3.496 | 43.814 | 463 | 958.85 | 1.48 | 1.47 | 0.95 | 0.04 | 1.43 | 0.46 | 90.11 ± 29.3 | 3.34 | 1.09 | 1.51 | 1.51 | 0.95 | 1.43 |
| Huttes top | 3.496 | 43.814 | 473 | 957.70 | 1.49 | 1.49 | 10.67 | 0.23 | 7.35 | 1.46 | 3.18 ± 0.64 | 4.80 | 0.96 | 0.69 | 0.69 | 10.67 | 7.35 |
| Leicasse | 3.558 | 43.817 | 245 | 984.16 | 1.22 | 1.22 | 9.9 | 0.21 | 44.55 | 1.87 | 22.92 ± 1.19 | 0.96 | 0.05 | 4.50 | 4.50 | 9.90 | 44.55 |
| Leicasse | 3.558 | 43.817 | 220 | 987.10 | 1.19 | 1.19 | 7.55 | 0.16 | 33.61 | 1.4 | 29.35 ± 1.51 | 1.00 | 0.05 | 4.45 | 4.45 | 7.55 | 33.61 |
| Leicasse | 3.558 | 43.817 | 240 | 984.75 | 1.21 | 1.21 | 1.44 | 0.04 | 1.32 | 0.4 | 28.8 ± 8.79 | 4.38 | 1.34 | 0.92 | 0.92 | 1.44 | 1.32 |
| Leicasse | 3.558 | 43.817 | 240 | 984.75 | 1.21 | 1.21 | 1.15 | 0.05 | 1.58 | 0.15 | 56.5 ± 6.03 | 3.53 | 0.38 | 1.37 | 1.37 | 1.15 | 1.58 |
| Leicasse | 3.558 | 43.817 | 240 | 984.75 | 1.21 | 1.21 | 0.46 | 0.04 | 1.13 | 0.33 | 267.19 ± 81.62 | 2.30 | 0.70 | 2.46 | 2.46 | 0.46 | 1.13 |
| Leicasse | 3.558 | 43.817 | 240 | 984.75 | 1.21 | 1.21 | 0.41 | 0.04 | 1.41 | 0.48 | 430.39 ± 152.7 | 1.58 | 0.56 | 3.44 | 3.44 | 0.41 | 1.41 |
| Leicasse | 3.558 | 43.817 | 240 | 984.75 | 1.21 | 1.21 | 1.13 | 0.04 | 1.79 | 0.6 | 67.16 ± 22.68 | 3.23 | 1.09 | 1.58 | 1.58 | 1.13 | 1.79 |

---

## Author Response (AR2)

Dear Reviewers and editorial team.

Thank you for the comments and remarks that helped increasing the quality of the paper. We hope our responses addressed the issues satisfactorily. Bellow is the point by point responses for both reviewers.
* * *
**RC1 (Fritz Schlunegger):**

*This paper still needs some technical improvements before it can be published. In particular, the results section reads like a method chapter, and the first chapter of the discussion section is actually the presentation of some of the results.*

Section 3 title is modified as "Methods" and title for section 4 modified as "Results and discussions". The final block from the initial section 3.2 is moved toward the beginning of the section 4.

**A section presenting all results in full detail, however, appears to be missing (concentrations of cosmogenic 10Be and 26Al, uncertainties, ages, denudation rates etc.). Figures 3 and 4 could be part of the results.** *I also miss a table in the main text where all results are shown.*
*I should have made this comment before, my apologizes to formulate this criticism so late in the review process.*

We choose to provide all the results directly in the form of the figures 3 and 4 and not to add the whole table in the main text because it would lead to a 2 to 3 pages-long table or selected information, hence being a reduced albeit duplicate of the supplementary one. We believe that providing all the information (sites parameters, concentration, chemistry, etc.) in one table is best for a possible latter-use and reproducibility.

*As a second criticism, there are very long sections (such as 4.2) without any reference to a figure or a table. Section 4.2 presents important information, but I don't see the related numbers of relationships on any figures. This would be important for me to verify if the statements are correct.*

References to Fig. 3 (line 220) Fig. 4 (line 231) have been added.

*As a third point, on p. 13, it is stated that the lower part of the Rocas cave has similarities to those reported for ghost-rock caves described before. However, the details are not presented. This is actually important, because the new model of cave formation bases on this comparison and also on the age pattern.*

We have added a sentence to describe what is similar to ghost-rock caves described before: "In this younger part of the cave, passages show morphologies similar to those reported for ghost-rock caves by Dubois et al. (2014) and Rodet (2014), that is to say the cross section of the galleries are characterized by lens (or half lens) shape extending in the weathered strata while the non-weathered strata and the lower part of the galleries are characterized by scallops, potholes and incised meanders. Furthermore, the preserved ghost-rock at the type of the lens shape have porosities larger than 10 % (Fig. 5).

Minor points:
*p. 2, line 41: it... physically erodes and transports insoluble sections. I think you mean: it transports particles that were eroded from insoluble sections. I think that this water does not transport entire sections (which would be too much).*

"[...] it also simultaneously physically erodes and transports insoluble sections of bedrock. "
modified as

"it also simultaneously physically erodes and transports insoluble residues from bedrock sections."

***p. 2, line 57: ..as cave sediment infill. Please avoid the juxtaposition of more than two substantives and change to: as sediment infill of a cave.***
"infill" modified as "deposits".

***p. 3, lines 82ff: The sentence starting with 'The latter has been used...' is not clear. Please rephrase and ev. make two sentences.***
"Large water flow loops at depth have been proposed to explain some hypogene cases since the flow is upward on one end of the loop (Klimchouck, 2017) or used to explain ghost-rock formation and its subsequent drain, sometimes creating a deep sump at more than 100m below the base level (Dandurand et al., 2019). The latter has been used to invoke convective cells as a satisfactory explanation for primokarst formation, subsequent drain, and finally deep phreatic loops such as Fontaine de Vaucluse or Touvre spring."
Modified as
"Large water flow loops at depth have been proposed to explain some hypogene cases since the flow is upward on one end of the loop (Klimchouck, 2017). Dandurand et al. (2019) refer to a similar process with large convection cells of water at depth to explain ghost-rock formation and its subsequent drain, sometimes creating a deep sump at more than 100m below the base level. In this model, deep convective cells are proposed as a satisfactory explanation for primokarst formation, subsequent drain, and finally deep phreatic loops such as Fontaine de Vaucluse or Touvre spring"

***p. 4, line 101: Inherited from what?***
"Inherited" modified as "Hercynian-inherited"

***p. 7, line 168: we use the Lal (1991) scaling factors***
Changed accordingly

***p. 7, line 173: erosion that provide modeled concentrations that... avoid the use of nested sub-sentences.***
"Both the minimal and maximal combination of burial age and erosion rate that provide modeled concentrations that are in the range of the measured one (± 1σ) are computed to estimate uncertainties."
Modified as
"Both the minimal and maximal combination of burial age and erosion rate providing modeled concentrations in the range of the measured one (± 1σ) are computed to estimate uncertainties."

***p. 8, line 178: We present in Figure 3 -> In Figure 3 we present***
Changed accordingly

***p. 10, lines 229ff: The related values have not been presented yet.***
A link the Fig. 4 is added.

***p. 10, line 247: Based on the cave locations -> please specify what you mean here.***
"Based on the cave locations in the vis River channel..."
modified as
"Because of the direct cave-entrances toward the Vis River channel, and the short distance (100s meters) between the sampling site and the cave entrances..."

**p. 12, line 287: that are distant of ... the formulation doesn't sound correct to me, but I might be wrong. It just sounds strange.**
We didn't change it as it seems ok, albeit probably not of a common usage.

**p. 14, line 330: rather than a river-related tiered cave.**
Changed accordingly

**p. 14, line 340: and that the evolution of the base level …**
Changed accordingly

**p. 15: The section of Figure 6 lists 4 items, but the figure itself shows only three sketches. Please harmonize, else it is quite difficult to properly understand this figure.**
The caption is modified and the 4 items meaning are integrated inside the figure itself
* * *
**RC3 (Régis Braucher):**

*The paper of Malcles et al. presents burial ages obtained in karstic networks of southern Massif Central. The authors propose for networks far from the river valley flanks or cliff walls that the well accepted epigenic speleogenesis model (network is formed when water table is stable then abandoned when the river is lower due to incision) cannot be applied and propose a model based on speleogenesis controlled by regressive denudation towards inner part of the plateau. Despite I am a bit far from this topic but more attracted by the cosmogenic nuclide applications, I think that these data have to be published after rewriting with more explanations and simplification. At this stage, some parts of the paper are a bit fuzzy, and the cosmogenic methodology lacks important information. See pdf*
Thank you, we hope we managed to make the purpose better explained ad that we did provide the cosmogenic-nuclide parameters that you found missing.

**Part 3 : Please provide the types of spikes used and their concentrations.**
We use in-house spike Abaz5870 with a concentration of 1025 µg/g. This information was added in the table caption.

**Precise the spallation production rate used.**
We 4.47 and 30.29 atm g$^{-1}$ yr$^{-1}$ as SLHL spallation production rate for $^{10}$Be and $^{26}$Al respectively. This was added line 161.

**What half-lives have been used for $^{10}$Be and $^{26}$Al?**
We used half-lives values from Korschineck (2010) and Chmeleff (2010): 1.387 and 0.705 Ma for $^{10}$Be and $^{26}$Al respectively. This was added line 124.

**What is the spallation production rate ratio used for $^{26}$Al/$^{10}$Be (6.75?)**
Yes, we use the value of 6.75. We indicate the total value SLHL instead (line 162 and associated references for both nucleonic and muonic productions)

**Line 109 – 123: the age calculation explanations are not clear and difficult to understand.**
We modified and completed the explanations, we hope it is clearer now.

*Using the data set provided I have recalculated all ages and paleo denudation rates (see excel table at the end of this review) using a normal approach sample by sample, ignoring postproduction. The clauside amalgam can be modelled (2.04 ± 0.46 Ma and 147.8 ± 33.16 m/Ma).*

Thanks for the recalculations, we also did sample-by-sample computations, but we don't know what you mean by "normal approach" here. The mostly insignificant discrepancy between both methods in term of burial ages comforts the proposal of this paper albeit the results for paleo-erosion rates seems sometimes statistically different (at one σ). Because only the order of magnitude should be considered for current river-sand estimation of recent denudation rate (e.g. Sassolas-Serrayet et al., 2019), and given the increased number of unknown for past conditions (source elevation, watershed morphology a few Ma ago, etc.), we do not consider this statistical discrepancy as meaningful and, as stated in the paper, we do consider only the order of magnitude as a useful indicator.

The assumption of no-secondary production for the Clauside site is wrong because of the too small overlying rock thickness with less than 10 m and probably a lower mean density due to alteration or fracturation of the overlying rock. This latter effect might be small. Indeed, we recognize that the secondary production during $10^5$ to $10^6$ yrs is reasonably smaller than the current one but because the time-production-rate path is not known we prefer not to provide any constraint that might be misleading for further user. A dedicated study using overlying carbonate denudation rate and high-resolution DEM could be performed in order to provide a sounded estimation.

**A banana plot will help to have in one figure the entire dataset.**

[Figure]

Thank you for the banana plot it shows that we provide all the information for anyone to reprocess our data. Unfortunately, since post-burial production cannot be neglected for some caves, as for example the Baume Clauside, we rather choose not to plot all the samples on one banana plot but provide all the data so anyone willing to have the approach followed by Régis Braucher can do it. We provide here the banana plot for Baume Clauside to illustrate that given the long burial of the sample and the rather low thickness of limestone above, it is impossible to give a constrained burial age.

[Figure]

Sample: CLAUSIDE_blk latitude= 43.96° elevation= 487m

*Regarding the production rate used in the calculation we do not know if it the one of the cave location or the one of the sources of the sediment (mean production rate of the watershed). This will not alter the burial ages but will highly influence the paleo denudation rate determination.*

Agree, we use the location of the sampling site (elevation, etc.) for the computation of the scaling factors. For the low-elevation cave with young samples, this assumption is probably wrong and one can assume that the paleo-watershed should have displayed the same kind of geometry and elevation than the current one (if no process as drainage capture or transient dam happened during this time). For older sample, the paleo-elevation or even mean latitude of the watershed is not known because of the regional dynamic (Massif-central uplift, Mediterranean watersheds being aggressors of the Atlantic ones, etc.). For these reasons we choose to use the sampling site parameters bringing at least a processing consistency.

"Theses scaling factors use the sampling site parameters (e.g. elevation)." was added line 170 to make it clear for the readers.

*Fig.3. try to use different symbols for a given site. This will help the reader working on black and white paper sheet. In this figure you have plotted two Rocas ages and two Fonctionnaire ages corresponding to two measurements on the same samples. If this is true do not present both data as this will give artificial more weigh to these ages. You can do this when working on different samples.*

We do not use the Rocas and the Fonctionnaire samples for incision rate computation, so they do not bring any artificial weight. We think that it is important to illustrate the repeatability of the measurements, and since they don't bring any artificial weighting we prefer to keep them. We have modified the symbols to help readers working on black and white paper sheets.

**Line 140-144: the use of isochron approach is not helpful here.**

We agree with the reviewer, but we think that we have to show the results of the isochron approach so the readers can make their own minds.

*Line 145-162: This par is hard to understand!! You are explaining that samples might have been already buried prior to they are deposited in the network; this yields to a scattering in the age distribution. How can you know the sample position in the alluvium cover before its burial in the network? (Line 52-155:" This sample with the younger age, was the one located closer to the surface in the surface deposited alluvium layer prior to burial. The older age (~4 Myrs) is a better measure, equal or younger, of the emplacement of the alluvium layer that was subsequently buried into the cave. This sample was the one located deeper in the surface alluvium layer before cave burial")*

Indeed, the main problem was the ~ 3 Ma of burial differences between the different cobbles in the Leicasse cave system. Because of their current location they do have a final burial in common.

First, this final burial stage can not be longer than the younger age (albeit it can take any value between this youngest age and "0").

If we consider the final burial period as equal to the youngest age (~ 1 Ma), it implies that this cobble did not endure burial prior to the final cave deposit, hence it stayed at the surface with an "infinite" exposition. Consequently, if the true final burial period is shorter than 1 Ma, it means that this cobble was partially buried close from the surface in an alluvium layer before its final burial in the cave. The true depth and residence time can not be properly estimated though.

Second, the total burial (subsurface partial or complete burial + cave) has to be equal or greater than the oldest age.

Consequently, this "4 Ma old cobble" was buried more than the "1 Ma old one". The logical explanation, using a parsimonious approach, is to consider a larger initial depth for the "4 Ma old cobble". But this cobble could have been also partially buried, although deeper that the youngest buried cobble, and therefore it represents a lower estimation of the age of the emplacement of the alluvium surface layer priori to its burial in the cave.

We added, line 246:

"We point out that theses point are relatives, that is to say, if it seems logical that the oldest being initially the deepest and the youngest the shallower, the absolute depth prior to the final burial, however, is unknown. A few constraints can be brought by the fact that the

*Line 179-180: What is the mean displacement rate of the CFZ fault, and the mean offset after earthquakes? In Ritz et al. one can find max offset values of 20 cm and it is also mentioned in the same paper that no surface deformation was observed during historical seism. Can you thus conclude that this fault can be responsible of the incision of the studied valleys? What about a global uplift due to Massif Central Mountains?*

We do think that the incision is permitted by the Massif-Central uplift and that the CFZ is a key element permitting a rather strong localization of the differential uplift, hence of the deformation. However, if the CFZ dynamic or the precise regional/local uplift rate are interesting questions, they are far out of the scope of this paper and our data only point toward a difference in incision rates, and are only supported by a few points.

Going further toward CFZ dynamic wouldn't be properly supported. For instance, prior to being able to discuss a hypothetical mean offset, the CFZ activity should be thoroughly demonstrated for other parts and shorter time scales. Then, it would only provide informations relative to a more or less constant differential uplift but not toward the repartition of this offset on different faults, or the proper rheology of such a system (fault locking, creeping, lateral variations etc.).

Therefore our aim was only to point out that our data suggest at least a gradient of incision rates across the CFZ. But we are unable to say if this is a regional tilt or a localized deformation on the fault. Given the Teil earthquake on the north-eastern end of the CFZ, proper studies should be conducted in our area too given our results.

*Fig. 4; change symbols and change police type for network far from the river cliffs.*

Changed accordingly, consistently with the Fig. 3

***Line 188: What do you mean by "The unexpected result of diminished burial ages shown in Figure 3…"?***

Our point here is that, given the elevation of the Larzac caves relatively to the river, and given the regional trend, we expected ages older than the obtained one (e.g. ~ 3 Ma for the Rocas samples). We added; line 291:

"(when compared with the expected one using the regional trend of ~ 90 m Ma$^{-1}$)".

***Line 198. Can you explain you approach here:" speleogenesis paradigm (ESP) which would predict ages 2 to 4 Ma older - or alternatively, a cave level elevation 150 to 250 m lower than recorded compared to the regional base level at the; me of the deposit)?***

Given the ESP, and without any ad-hoc complications due to paragenetism, etc. the age-elevation relationship is expected to follow a more or less regular trend: the higher the deposit, the older the age. When using the regional ~ 90 m Ma$^{-1}$ of incision rate we can predict an age for the Rocas or Leicasse, etc. deposit. This predicted age (~ 3 Ma) is way too old when compared to the obtained one (~ 1 Ma).

***Line 202: Why the absence of sediment in Rocas implies an age younger than 1 Ma?***

This model is in our opinion the best one that can explain the data without the need of many ad-hoc assumptions or physically unsounded hypothesis. Given the fact that the sedimentary infilling dated at 1Ma are incised, at least one erosional phase is needed after the deposition. Because there is no quartz infilings to be found in the lower part of the Rocas, we assume that this lower network did not exist when the 1 Ma old sediments settled in the upper part. Indeed, if the lower part existed at that time, it is reasonable to think that quartz could be found somewhere (e.g. hydrological shadow areas).

It is indeed possible to imagine other models with the lower network being present prior to 1 Ma (or else), as for example assuming a total infilling of the network followed by a total removal of all the quartz in the lower parts only, etc. but such models quickly tends to be irrefutable and physically complex (where did all the quartz, in terms of volume, came from and where did it goes, etc.). Therefore we prefer the simpler model with formation post 1 Ma.

***Line 208: Scorpions and Bergougnous sites seem to be affected by the Vis River. Why do you compare the Rocas sediments (from alluvial deposits on top of the plateau) with these two sites?***

We compare the Scorpions/Bergougnous and the Rocas to highlight the inadequacy of the ESP relatively to the age-elevation relationship. At first, and along the ESP model, we assumed that the Rocas (or Fonctionnaire, Leicasse) would have shown older ages than the caves located lower (Scorpions, Escoutet, etc.).

***Why the same age of 1 Ma cannot be related to the activity of the entire network from Sc/Be to Rocas?***

See answer to the "Line 202" question.

***As you proposed a new formation model it is worth better explaining this last part synthetized by fig. 6 and show how you construct the chronology from 1 Ma to present.***

Thanks for saying that we propose a new formation model, but this not true, it was already observed in near real time by Yves Quinif and collaborators as well as by Joël Rodet based on field observations. We just observed it at a larger scale using TCN, which wasn't our goal since we were aiming at constraining incision rates in the area. To make it more clear that it is related to the downcutting of the canyon in the limestone plateau and hopefully make it more clear for the readers we have modified figure 6.

---

## Author Response (AR3)

**Editors Comment (Richard Gloaguen):**

**After careful review of the reviewers' comments and the answers provided by the authors I tend to lean accepting this manuscript at the condition that the authors add a paragraph in their discussion in which they acknowledge the issues / alternative interpretations raised by the reviewers.**
**I think that, at this stage, it is more a matter of interpretation than methodological or analytical problems.**
**My point is, this submission should be published as it triggers discussion and the points mentioned by the reviewers will be available online. Nonetheless to make it clear that there is matter of contention, the authors should add a short discussion indicating that some interpretations need further validation/ additional data.**

*Authors answer:*

*Dear editor.*

*We are pleased that, pending the above-mentioned required modifications, you have accepted to publish this manuscript. We thank you for this decision and we hope our answer is satisfactory. The discussions and important interrogations regarding the interpretations that arose during the review, does indeed call for further investigations in order to try to refute, following Karl Popper, the different opinions and paradigms that were expressed relatively to the karst structuration and relationship with the ghost-rock phenomenon. In this way, we thank the reviewers for their remarks that helped us to take into consideration other points of views and we hope our contribution provides original data and advances regarding the knowledge of the karst morphology and dynamic.*
*We added at the end of the discussion:*
*"As pointed out by Dubois et al. (2022) karst morphologies are used by scientists to speculate on processes that induce speleogenesis. It leads to a tremendous number of different processes to form caves (see for example Figure 3 of Harmand et al., 2017). Here we choose to follow an approach driven by the principle of parsimony also termed as Ockham's razor and propose a continuum process where cave geometry complexity is only driven by the primary phase of alteration. We are not the first authors to do so (e.g. Dubois et al, 2014, 2022), but acknowledge that this is an ongoing debate as attested by the discussions with the reviewers triggered by the first drafts of this study. Quinif (2010) suggested the need for a new paradigm about karstogenesis implying ghost-rock processes. We know from the history of sciences that shifting from one paradigm to another is a complex journey (Kuhn, 1962). More studies and debates will be needed to overcome the present matter of contention about how ghost-rock processes should be considered in karstogenesis, that is to say, rather as a secondary process (e.g., Schmidt, 1974, Klimchouk, 2012) or the primary process (e.g., Rodet, 2014, Dubois et al., 2014 and this present study)."*